# Alveolar Ridge Preservation Using a Novel Species-Specific Collagen-Enriched Deproteinized Bovine Bone Mineral: Histological Evaluation of a Prospective Case Series

**DOI:** 10.3390/bioengineering11070665

**Published:** 2024-06-28

**Authors:** Andreas van Orten, Werner Goetz, Hakan Bilhan

**Affiliations:** 1Private Dental Practice Do24, Dortmunder Str. 24–28, 45731 Waltrop, Germany; andreas.van.orten@icloud.com; 2Policlinic of Orthodontics, Centre for Dental Care, Basic Science Research in Oral Biology, Friedrich-Wilhelms University, Welschnonnenstr. 17, 53111 Bonn, Germany; wgoetz@uni-bonn.de; 3Department of Periodontology, School for Health Sciences, Witten/Herdecke University, Alfred-Herrhausen-Str. 45, 58448 Witten, Germany

**Keywords:** extraction socket, alveolar ridge preservation, socket preservation, bone substitute material, bone graft, xenograft, cross-linked collagen membrane, dental implant, socket healing, deproteinized bovine bone mineral, guided bone regeneration

## Abstract

In recent years, the significance of maintaining the alveolar ridge following tooth extractions has markedly increased. Alveolar ridge preservation (ARP) is a commonly utilized technique and a variety of bone substitute materials and biologics are applied in different combinations. For this purpose, a histological evaluation and the clinical necessity of subsequent guided bone regeneration (GBR) in delayed implantations were investigated in a prospective case series after ARP with a novel deproteinized bovine bone material (95%) in combination with a species-specific collagen (5%) (C-DBBM). Notably, block-form bone substitutes without porcine collagen are limited, and moreover, the availability of histological data on this material remains limited. Ten patients, each scheduled for tooth extraction and desiring future implantation, were included in this study. Following tooth extraction, ARP was performed using a block form of C-DBBM in conjunction with a double-folded bovine cross-linked collagen membrane (xCM). This membrane was openly exposed to the oral cavity and secured using a crisscross suture. After a healing period ranging from 130 to 319 days, guided trephine drilling was performed for implant insertion utilizing static computer-aided implant surgery (s-CAIS). Cores harvested from the area previously treated with ARP were histologically processed and examined. Guided bone regeneration (GBR) was not necessary for any of the implantations. Histological examination revealed the development of a lattice of cancellous bone trabeculae through appositional membranous osteogenesis at various stages surrounding C-DBBM granules as well as larger spongy or compact ossicles with minimal remnants. The clinical follow-up period ranged from 2.5 to 4.5 years, during which no biological or technical complications occurred. Within the limitations of this prospective case series, it can be concluded that ARP using this novel C-DBBM in combination with a bovine xCM could be a treatment option to avoid the need for subsequent GBR in delayed implantations with the opportunity of a bovine species-specific biomaterial chain.

## 1. Introduction

As reported in the World Health Organization’s (WHO) 2022 global oral health status report, approximately 3.5 billion individuals globally are affected by oral diseases, with dental decay being the most common among them, which affects approximately 2 billion adults and over 500 million children with primary tooth decay. Other common oral health issues include periodontal diseases, tooth loss, and oral cancers. Complete tooth loss affects around 7% of individuals over the age of 20 [1]. 

In general, tooth loss requires therapeutic intervention [2], and dental implant placement has been a reliable treatment option for several decades [3,4]. An adequate bone volume is essential for the long-term success of prosthetic restorations supported by implants [5]. Adequate bone volume can be achieved either before implantation or during the implant insertion procedure [4]. 

Minimally invasive or flapless techniques offer advantages for both practitioners and patients [6]. Having an adequately dimensioned bone volume during the implant placement procedure may enable immediate implant loading or reduce the treatment duration compared to implantation with guided bone regeneration (GBR) as the native bone can usually tolerate earlier functional loading compared to simultaneously augmented bone [7]. 

After tooth extraction, notable alterations take place in the adjacent tissues. The alveolar bone undergoes remodeling, with resorption outweighing new bone formation [8]. These processes occur simultaneously in overlapping phases and have been observed in both animal models [9] and humans [10,11,12]. Generally, horizontal dimension loss is more pronounced than vertical dimension loss. A significant portion of structural changes takes place within the first three months after extraction [8,13]. The weighted mean horizontal loss during this period is 3.87 mm, while the vertical loss is 1.67 mm [14]. Resorption is more pronounced on the vestibular side than the oral side and is greater in the crestal areas than apically [8]. The extent of bone loss can be particularly significant in the esthetic zone, where the bundle bone over the tooth roots can be exceptionally thin [15]. This phenomenon is often masked by a rise in soft tissue thickness following tooth extraction, making the purely clinical evaluation in the esthetic zone misleading [16]. The magnitude of the subsequent bone loss depends on various factors, largely based on experiences from guided bone regeneration (GBR) and periodontal bone regeneration, such as the number of remaining bone walls, site-specific space maintenance, stabilization of the blood clot, and the extent of trauma during tooth extraction, all of which can influence the volume of bone loss [17].

Different techniques aiming to limit or compensate for the extent of volume loss have been described.

Alveolar ridge preservation (ARP): The extraction socket is packed with a biomaterial using this technique. It can be performed with or without the use of resorbable or non-resorbable membranes. When primary wound closure is desired, the technique resembles GBR. However, primary wound closure may have disadvantages, such as negative effects on the local blood supply, recession and papilla defects in adjacent teeth, shifting of the mucogingival border, a loss of keratinized mucosa, swelling, hematoma, and increased pain. ARP is generally unable to completely prevent resorption [18,19].

Socket seal technique (SS): First described almost 30 years ago [20], this technique focuses primarily on sealing the socket using autogenous soft tissue or membranes. By avoiding the use of autogenous soft tissue, benefits such as reduced morbidity at the donor site and decreased scar formation in the previous socket area can be achieved [21]. This technique can be performed with or without additional socket filling using biomaterials. A variation of this technique involves sealing the socket with a root slice [22]. SS also does not typically prevent resorption entirely [23,24].

Socket-shield technique (SST), modified socket-shield technique, partial extraction therapy (PET): Preserving root fragments and maintaining the periodontal ligament along with its associated blood supply can help preserve the bundle bone in these areas even after removing the remaining tooth structure [25]. Originally, this technique was described in combination with immediate implantation [26] as the technique of immediate implantation itself is not suitable for preventing bone resorptive processes [27]. The modified socket-shield technique [28] can be considered a variation of the SST, where implant placement is delayed. It is crucial to take measures after removing the tooth fragment to prevent epithelial ingrowth and facilitate bone regeneration [29]. The modified SST combines aspects of the SST and ARP. Preserving suitable intact root fragments utilizing the PET, such as in the areas of future pontics, cantilevers, or future implant sites, is also effective in minimizing the extent of resorption [30]. 

Orthodontic and extrusion techniques: By using orthodontic appliances, teeth or portions of teeth can be selectively and time-delayed extracted or exfoliated, allowing targeted bone and soft tissue regeneration at future implant sites, thus avoiding or minimizing the need for GBR during implantation [31]. 

Biomaterials: A variety of materials are utilized in ARP or SST, often in combination [17]. These include bone or bone substitute materials (autogenous, allogeneic [32], xenogeneic, phylogenetic), tooth fragments (autogenous [33], xenogeneic), alloplastic materials (e.g., calcium sulfate, beta-tricalcium phosphate, hydroxyapatite, bioactive glasses), and biologics (e.g., platelet-rich fibrin, platelet-rich plasma, hyaluronic acid, recombinant human bone morphogenetic protein-2, enamel matrix derivatives), potentially resulting in better histomorphometric outcomes and quicker wound healing compared to control groups [34]. 

The different bone substitute materials vary in their degradation behavior. In general, it can be said that allogeneic bone is resorbed more rapidly than porcine bone, and porcine bone is resorbed faster than bovine bone. Conversely, the volume of newly formed bone after augmentation typically follows the opposite pattern. Among the alloplastic materials, some undergo minimal or no degradation (e.g., bioactive glasses), while others are more resorbable (e.g., β-TCP), and some are rapidly resorbable (e.g., calcium sulfate) [35]. By combining various bone substitute materials and biologics, specific properties can be achieved [17]. For many patients, ideological factors (e.g., veganism, vegetarianism) or religious and moral considerations play an important role in the selection of a suitable bone substitute material [36]. Some bone substitute materials are available in block form, combined with collagen, to simplify handling compared to the use of granules. 

There is currently a research gap in the scientific literature regarding species-specific bovine bone substitute materials in block form. The purity of species is globally significant for many patients due to religious reasons. Introducing a novel material chain could potentially offer a straightforward and consistent therapeutic option for both practitioners and patients. 

This investigation focuses on a newly introduced deproteinized bovine bone material (95%) combined with species-specific collagen (5%) (C-DBBM) on the market. This combination, formulated into a block form, is utilized for alveolar ridge preservation (ARP) along with a double-folded bovine cross-linked collagen membrane (xCM). 

In this small prospective case series, the clinical and histological occurrence of anomalies and issues, as well as the clinical performance of this material chain, will be examined.

## 2. Materials and Methods 

Between September 2018 and October 2020, patients were enrolled in a private dental practice in Germany to take part in this prospective case series (Table 1). The inclusion criteria were as follows: age above 18 years, preference for delayed implantation rather than immediate, no medical history contraindicating the surgical procedure, and the planned extraction of a tooth classified as socket type I [37]. They had no peri-implantitis and no periodontitis or Stage I and Grade A periodontitis. All the patients were participating in regular recalls with semi-annual clinical check-ups and oral hygiene instructions.

The exclusion criteria included systemic diseases that could affect bone metabolism, antiresorptive therapy (such as bisphosphonates), pregnancy and lactation periods, psychiatric conditions, and oncologically significant diseases. Smokers and patients with diabetes mellitus were not excluded. One patient was an active smoker, consuming 10 cigarettes per day. Four patients were former smokers, and five were non-smokers. The study group included two hyperglycemic patients (Patient #4, a 77-year-old female with HbA1c = 6.5%; Patient #6, a 54-year-old male with HbA1c = 6.5%). Each participant provided written informed consent to join the study. The treatment followed the standard protocol of this dental practice, approved by the Ethics Committee of the University of Bonn (ethical committee decision #222/05). All the procedures and follow-up examinations were conducted by the same practitioner with 20 years of experience in oral surgery. 

The teeth required extraction for various reasons, but a consistent ARP protocol was utilized for all the sockets. Multirooted teeth were sectioned with a small rotating Lindemann bur (H162AZ, Komet, Gebr. Brasseler GmbH; Lemgo, Germany), and the root fragments were carefully and atraumatically elevated and extracted using matching periotomes (PT Periotomes, Hu Friedy, Chicago, IL, USA) under local anesthesia (Ultracain DS forte, Sanofi, Paris, France) to minimize the risk of iatrogenic damage to the alveolus. The remaining granulation tissue was removed using degranulation burs (EthOss, Ethoss Regeneration Ltd., Silsden, UK) (Figure 1) to prevent any adverse effects on alveolar healing due to soft tissue remnants. The biomaterials used included a native bovine, cross-linked collagen membrane (xCM) composed primarily of type 1 collagen fibers from the Achilles tendon (Memlok RCM, BioHorizons, Birmingham, AL, USA) and a deproteinized bovine bone material combined with species-specific collagen (C-DBBM) in the form of moldable bone blocks (MinerOss X Collagen, BioHorizons, Birmingham, AL, USA) (Figure 2), in accordance with the EFP recommendations at that time [18] and rehydrated according to the manufacturer’s instructions. The bone substitute material is predominantly composed of 95% anorganic cancellous bovine mineral granules and 5% bovine collagen. It undergoes terminal sterilization and exhibits a particle size ranging from 250 to 1000 µm. The bony edges of the sockets were exposed using a periosteal elevator. The xCM was carefully adapted to the sockets, ensuring an overlap of 2 mm to prevent soft tissue ingrowth. Subsequently, the vestibular part of the membrane was repositioned for direct access to the sockets. The C-DBBM blocks were cut to the appropriate size with a 15C scalpel. After placement into the sockets, moderate compression was applied with a bone compactor, avoiding overfilling (Figure 3). The membrane was double-folded on the oral aspect and repositioned over the bony edges of the sockets. The membranes were left exposed to the oral cavity. Complete wound closure, which would require extensive mobilization and potentially shift the mucogingival border, was not attempted. A crisscross suture using a Glycolon 6-0 resorbable thread (Resorba Medical GmbH, Nurnberg, Germany) provided temporary stability and slight wound margin adaptation. If necessary, papillae adaptation was performed using the same suture material with single interrupted sutures (Figure 4). A postoperative dental X-ray was taken to document the completion of the ARP procedure (Figure 5). The provisional treatment involved clip-anchored removable dental prostheses to prevent elongation, tooth tilting, or psychosocial issues due to tooth gaps, ensuring proper food intake. The patient was given non-steroidal anti-inflammatory drugs for pain relief (600 mg Ibuprofen, Ibuflam, Zentiva, Pharma GmbH, Berlin, Germany). The postoperative regimen included avoiding mechanical plaque control in the treated area for one week and using an alcohol-free chlorhexidine mouth rinse 0.2% (Chlorhexamed, GlaxoSmithKline Consumer Healthcare GmbH & Co. KG, Munich, Germany) twice daily. The sutures were removed after four weeks. A healing period of at least 19 weeks was planned to ensure a stable implant site. Presurgical assessment of the alveolar ridge was performed using cone beam computed tomography (Orthophos XG 3D, Dentsply Sirona, York, PA, USA) to enable flapless surgery by fabricating a static computer-aided implant surgery drilling guide. After administering local anesthesia (Ultracain DS forte, Sanofi, Paris, France), the implant site was prepared with a mid-crestal incision to maintain sufficient keratinized tissues. The implant site was prepared with a trephine bur (Trephine Ejection Kit, Hager & Meisinger GmbH, Neuss, Germany), the intended implant was placed, and a postoperative radiographic examination was performed (Figure 6). After carefully removing the bone core from the trephine, it was stored in a buffered 10% formalin solution. The implants healed submerged. The sutures were removed after ten days. The implants were uncovered after four months, followed by impression-taking after another four weeks and the final placement of screw-retained single-tooth restorations, consisting of custom titanium abutments with lithium disilicate crowns. A dental X-ray was taken at the conclusion of the prosthetic phase (Figure 7).

### 2.1. Histology

Each biopsy sample was immersed in 4% buffered formaldehyde (Sörensen buffer) at room temperature (RT) for at least one day for fixation, followed by decalcification in 4.1% disodium ethylenediaminetetraacetic acid (EDTA) solution for approximately 2 to 3 weeks. Post-hydration, the tissues were dehydrated using an ascending series of ethanol and subsequently embedded in paraffin. Serial longitudinal sections of 2–3 μm thickness were prepared, and representative slides were stained with hematoxylin-eosin (HE) and periodic acid-Schiff (PAS) reaction. To identify osteoclasts, the selected tissue sections underwent staining for tartrate-resistant acid phosphatase (TRAP).

### 2.2. Immunohistochemistry

Representative slides from the median sections of the sample series were deparaffinized, rehydrated, and rinsed for 10 min in Tris-buffered saline (TBS). Endogenous peroxidase was blocked by immersing the slides in a methanol/H_2_O_2_ solution (Merck, Darmstadt, Germany) for 45 min in the dark. The sections were then pretreated with PBS containing 1% bovine serum albumin (BSA) for 20 min at room temperature, digested with 0.4% pepsin for 10 min at 37 °C, and incubated with primary antibodies in a humid chamber. The following markers were examined: osteocalcin (OC), collagen type I, and von Willebrand factor (vWF). Details of the antibodies and incubation protocols are listed in Table 2. Antibody binding was detected using the peroxidase-conjugated EnVision^®^ anti-mouse system or the EnVision^®^ anti-rabbit/anti-goat HRP-conjugated secondary antibodies (Dako, Glostrup, Denmark), diluted 1:50 and incubated for 30 min at room temperature. Peroxidase activity was visualized using diaminobenzidine (DAB), producing a brown staining product, and the slides were counterstained using Mayer’s hematoxylin. Specificity controls were conducted by (i) omitting primary antibodies and using TBS or normal horse serum instead or (ii) omitting either primary or secondary antibodies. Mandibular bone or fetal human bone tissues known to carry the antigens were used as positive controls.

### 2.3. Histological Evaluation

Stained sections were examined and assessed using a light microscope (Zeiss, Jena, Germany), with images captured digitally via an integrated camera.

## 3. Results

Ten subjects, with an average age of 53 years (ranging from 42 to 77, SD 11.5 years), including five males and five females, were enrolled in this case series. They presented with 10 prospective implant sites (following alveolar ridge preservation procedures) and underwent the prescribed protocol. In total, four premolars and six molars were extracted for the following reasons: poor endodontic prognosis (*n* = 5), unrestorable condition (*n* = 2), vertical root fracture (*n* = 2), and severe tooth tilting into a tooth gap (*n* = 1). In no instance did the procedure necessitate additional guided bone regeneration (GBR). Macroscopic clinical evaluation revealed an uneventful wound-healing process in all the patients.

The follow-up period ranged from 2.5 to 4.5 years after the initial surgery and from 2 to 4 years post-prosthetic loading. All the sites were restored using screw-retained suprastructures with custom titanium abutments and extraorally bonded lithium disilicate crowns. The implant survival rate was 100%, and all the implants were classified under implant quality scale group I according to the implant success criteria established at the PISA consensus conference. No technical or biological complications were reported. Regular recalls, including semi-annual clinical check-ups and oral hygiene instructions, were conducted for all the patients.

The gender distribution showed a higher proportion of male patients (60%) compared to female patients (40%). The average age of the patients was 53 years, with a range from 42 to 77 years. Most patients were between 44.25 and 60.25 years old, as reflected by the interquartile range.

The most common reasons for tooth loss were endodontic problems (50%), followed by unrestorable teeth (20%), fractures (20%), and tooth tilting (10%). The integration periods of the graft materials varied significantly, ranging from 130 to 319 days, with an average of 204.1 days.

Regarding smoking behavior, half of the patients (50%) never were smokers, while 40% were former smokers and 10% were current smokers (with a consumption of 10 cigarettes per day). Concerning general medical history, most patients (60%) had no abnormal medical history. Two patients had medicated type 2 diabetes, and two patients were known to be atopic individuals.

Under low magnification, most biopsies appeared as cylindrical specimens primarily composed of cancellous bone, featuring interconnected trabeculae of varying diameters, MinerOssX (MOX) granules with active osteogenesis, and intertrabecular connective tissue (Figure 8, Figure 9 and Figure 10). Areas of osteogenesis were mainly localized to the more coronal parts of the biopsies (Figure 8). Artificial ecchymosis and fragmentation of bone or connective tissue due to trephination were evident in nearly all the specimens. The basophilic MOX granules varied in size and shape, showing signs of degradation (Figure 9, Figure 10, Figure 11 and Figure 12). All the biopsies exhibited a network of cancellous bony trabeculae formed through appositional membranous osteogenesis at various stages around MOX granules or larger spongy or compact ossicles with minor granule remnants (Figure 9 and Figure 10). Newly formed bone was fibrous (Figure 9 and Figure 10). The early stages of perigranular osteogenesis were marked by the ingrowth of connective tissue into granules and the formation of osteoids around them (Figure 11). However, nearly all the specimens contained MOX granules without signs of osteogenesis, most displaying a thin, peripheral, dark basophilic layer (Figure 12). In some instances, this layer was visible as an interface between the granules and newly formed bone. Some surfaces of newly formed bone were covered by osteoblasts (Figure 13). In some specimens, the fibrous bone had already remodeled into mature cancellous or compact bone, appearing as lamellar bone with incorporated fibrous bone remnants (Figure 13). The bone surfaces were mostly covered by lining cells. The intertrabecular tissue consisted of loose or fibrous connective tissue with fibroblasts and moderate vascularization (Figure 10, Figure 11 and Figure 12). Osteoclasts were present on the surfaces of the newly formed bone and MOX granules (Figure 13). No foreign body giant cells were detected. Small, loosely arranged infiltrations were observed in three specimens (Figure 14). Necrosis was not observed. TRAP-positive osteoclasts were found on the surfaces of MOX granules without osteogenesis or bone deposition (Figure 15). COL I immunostaining showed weak to moderate staining in the matrix of newly formed bone, with stronger staining in osteoid seams and osteocytes. Focal immunoreactivity was observed in osteoblasts (Figure 16). Additionally, connective tissue staining was present. COL I exhibited weak to moderate staining intensity in the granule matrix and their interfaces (Figure 16). The newly formed bone matrix showed weak OC immunostaining, with most osteoblasts, osteocytes, and some fibroblasts near the bone surfaces staining more intensely (Figure 17). The interfaces between the MOX particles and newly formed bone and granule matrix were also reactive. Staining for vWF indicated moderate to good vessel density in most specimens, with capillaries, small arterioles, and large sinusoids being the predominant vessel types located between the bone trabeculae and granules (Figure 18). There was no clear correlation between the progression of osteogenesis and the duration of bone substitute placement. In patients with diabetes and atopy, osteogenesis was poorly developed.

## 4. Discussion

The objectives of this study were to assess the clinical utility and histological outcomes of alveolar ridge preservation using a novel species-specific C-DBBM in conjunction with a bovine double-folded xCM. The inclusion criteria for the patients were broad, while the exclusion criteria were limited to essential factors. This facilitated a diverse patient group intended to closely resemble clinical reality. However, due to the very low total number of patients and sites, a statistically representative depiction and derivation of universally applicable conclusions are either not feasible or greatly limited. C-DBBM has been known for some time to statistically significantly counteract horizontal and vertical shrinkage during alveolar ridge preservation [17,38]. ARP can potentially decrease horizontal bone resorption by 1.99 mm and vertical bone resorption by 1.72 mm [39,40]. In this case series, GBR was not necessary in any case. Although soft tissue grafting was not necessary in this case series, Seyssens et al. demonstrated that simultaneous soft tissue grafting using ARP is likely insufficient to fully compensate for the remodeling effects following extraction and that the invasiveness of the procedure is generally expected to be lower at a later time point compared to simultaneous ARP and soft tissue augmentation [40]. In this series of cases, maintaining the alveolar ridge dimensions was sufficient for uncomplicated, straightforward implantations. The absence of the need for GBR in this case series may have also been influenced by the location of the implant insertion sites in the premolar and molar regions and the requirement that the patients meet the inclusion criteria for socket type 1 according to Elian [37]. Nevertheless, these data align with the findings of Mardas et al., who estimated the need for additional GBR at 0–15% [41]. When interpreting the results of this study, several limitations must be considered. First, the lack of a control group without any intervention makes it difficult to quantify the absolute benefits of the technique employed. The sample size of this study, with a case number of 10, only allows for an initial, limited insight into the topic. Furthermore, patient selection limits the generalizability of this study. Another weakness of this study is the heterogeneity in the time duration from ARP to implantation. Due to the COVID-19 lockdowns and patient uncertainty during the COVID-19 pandemic, the variability in this factor was very high. An additional notable constraint in this study is the lack of histomorphometric analyses, so precise statements regarding the rate of new bone volume cannot be made. Uncertainty regarding the findings also exists concerning long-term outcomes, such as implant survival rates and long-term implant success, as data for this specific material combination have not yet been collected at this time. Further potential sources of error in addition to patient selection errors in this study may include:−Surgical technique and sample collection: However, it is considered simple, and the risk is rather low.−Laboratory processing and histological evaluation: These were conducted by a highly experienced team with decades of experience, and the risk in this case is also considered low.

## 5. Conclusions

Within the limitations of this case series, with clinical and histological evaluation after ARP using a novel species-specific C-DBBM, it was able to prevent the need for GBR in all the cases in the context of late implantation in patients with socket type 1 according to Elian. This could provide an additional valid treatment option for future patients seeking a purely bovine material chain, especially for block-form bone substitute materials. Clinical and histological follow-up examinations of the patients revealed no adverse, remarkable, or adverse reactions. However, further studies using significantly larger sample sizes and control groups, greater homogeneity in terms of the duration of bone substitute material retention after ARP and before implantation, and a more standardized patient selection with longer follow-up periods and histomorphometric analyses are desirable and necessary to confirm these results.

## Figures and Tables

**Figure 1 bioengineering-11-00665-f001:**
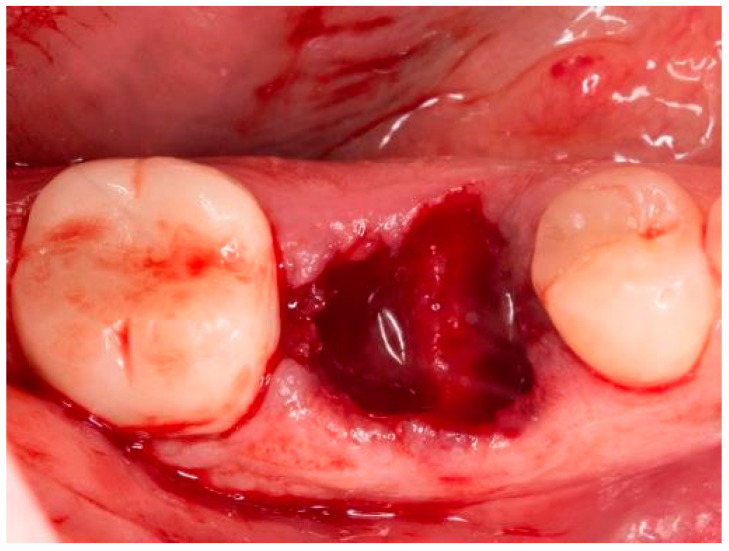
Socket of tooth 46 after tooth extraction.

**Figure 2 bioengineering-11-00665-f002:**
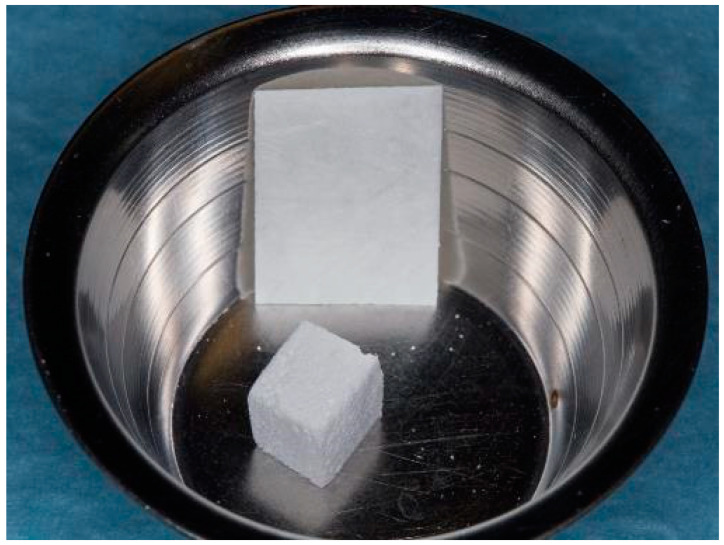
Bone graft material and membrane before rehydration.

**Figure 3 bioengineering-11-00665-f003:**
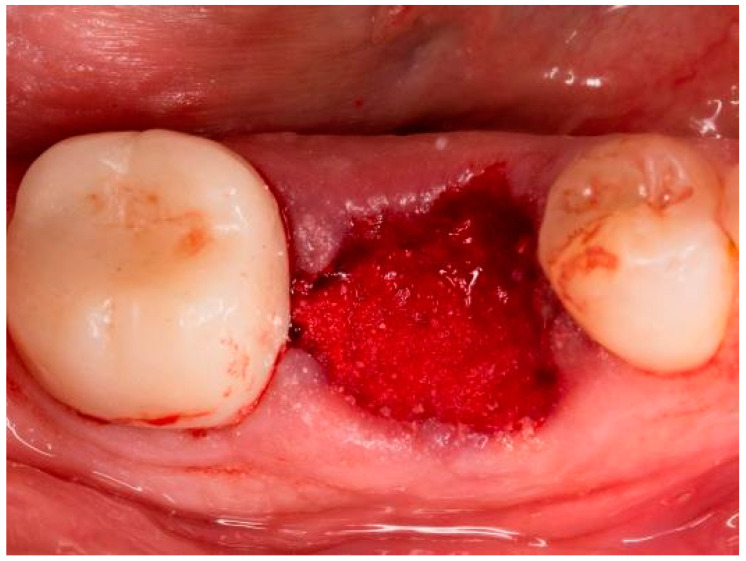
Socket of tooth 46 after bone graft placement.

**Figure 4 bioengineering-11-00665-f004:**
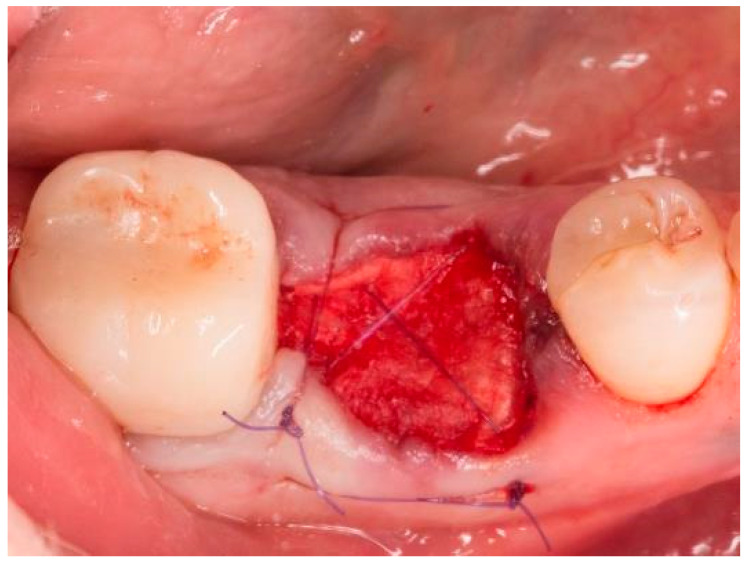
Socket of tooth 46 after suturing.

**Figure 5 bioengineering-11-00665-f005:**
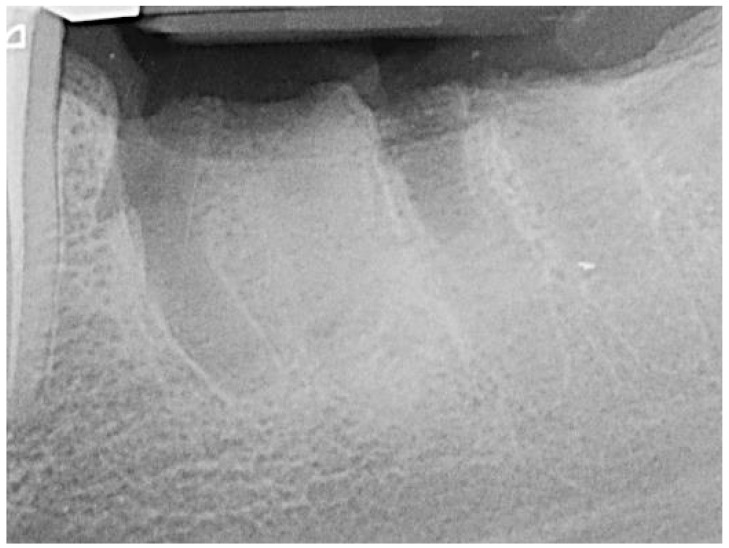
Socket 36 after alveolar ridge preservation.

**Figure 6 bioengineering-11-00665-f006:**
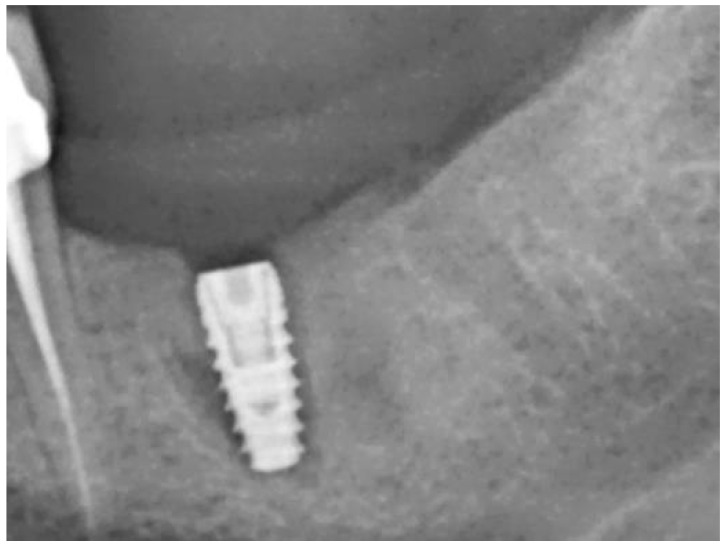
Regio 36 after implantation.

**Figure 7 bioengineering-11-00665-f007:**
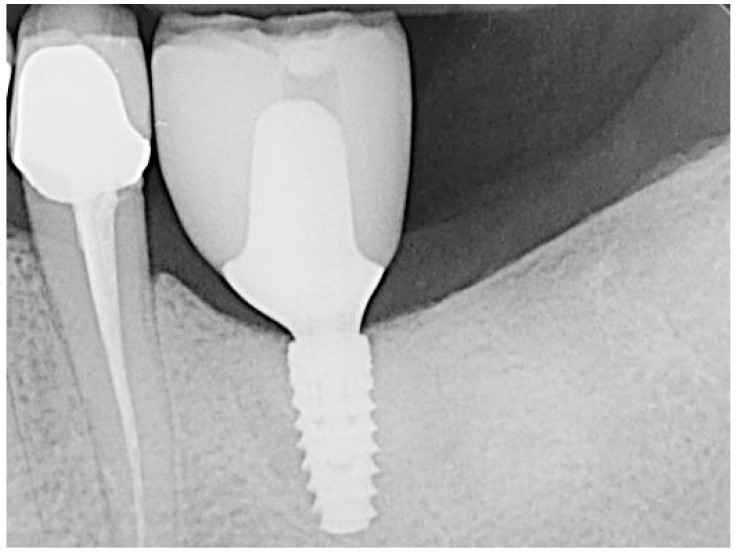
Regio 36 after prosthetic treatment.

**Figure 8 bioengineering-11-00665-f008:**
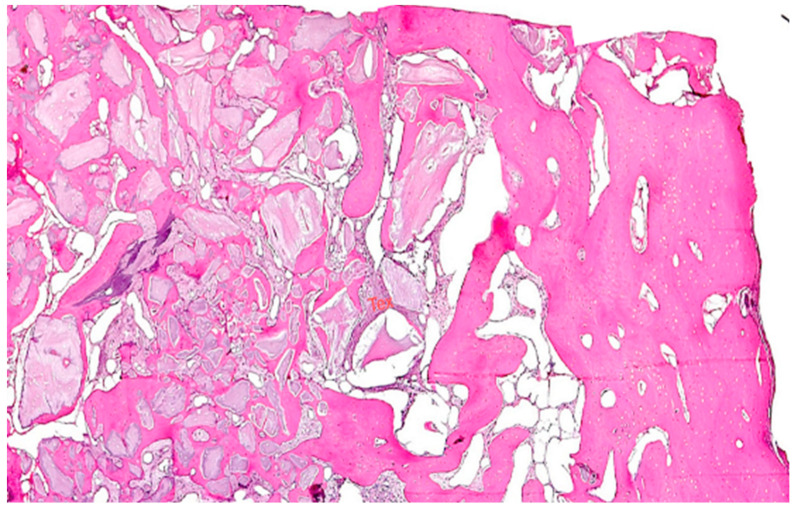
Reconstruction biopsy case 6, apical region on the right, coronal region with osteogenesis around bone graft granules on the left, HE, original magnification ×5.

**Figure 9 bioengineering-11-00665-f009:**
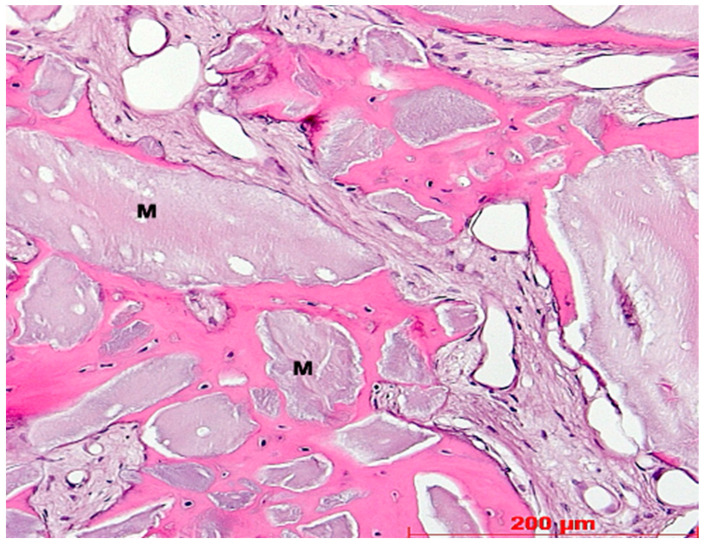
Biopsy case 6, perigranular osteogenesis around bone graft granules (M), HE, original magnification ×20.

**Figure 10 bioengineering-11-00665-f010:**
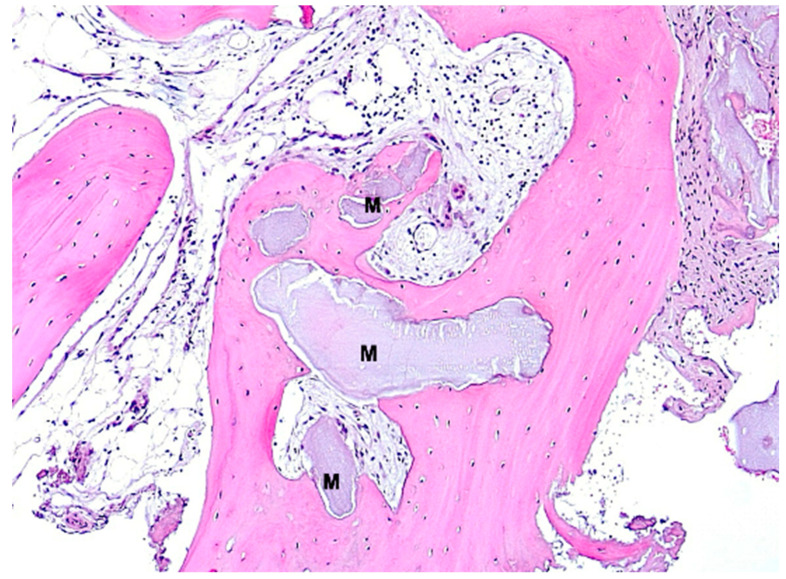
Biopsy case 8, advanced osteogenesis around bone graft granule remnants (M), HE, original magnification ×10.

**Figure 11 bioengineering-11-00665-f011:**
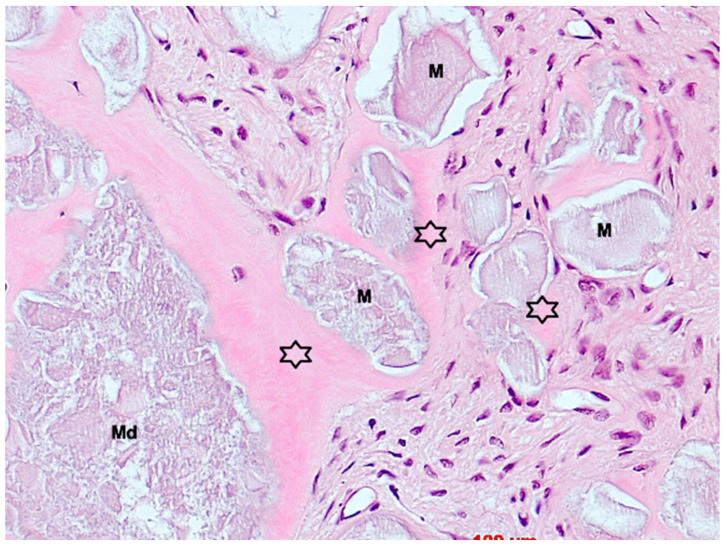
Biopsy case 9, early stage of osteogenesis by perigranular osteoid formation (asterisks), M = bone graft granules, Md = degraded bone graft granule, HE, original magnification ×40.

**Figure 12 bioengineering-11-00665-f012:**
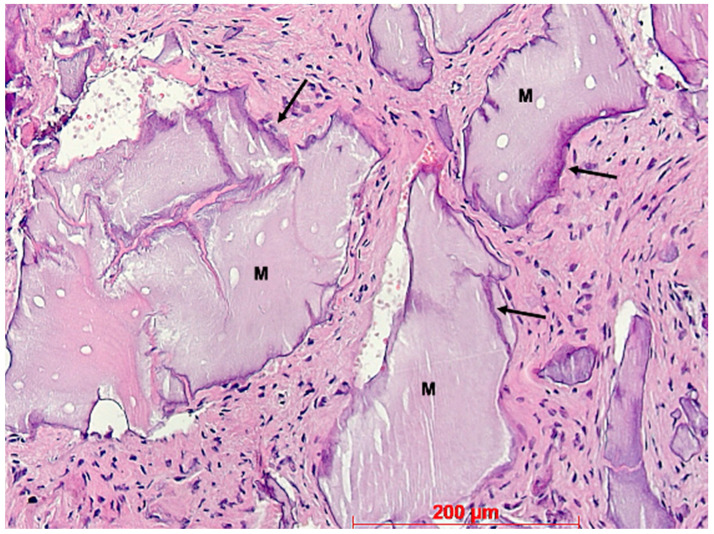
Biopsy case 4, bone graft granules (M) without osteogenesis and basophilic interfaces (arrows), HE, original magnification ×20.

**Figure 13 bioengineering-11-00665-f013:**
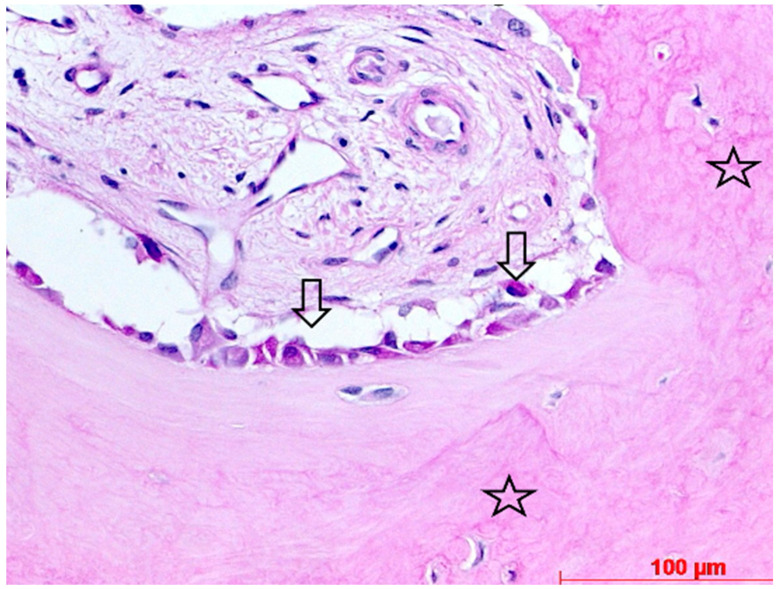
Biopsy case 1, osteoblasts on the surface of newly formed bone (open arrows), stars = fibrous bone, PAS, original magnification, ×40.

**Figure 14 bioengineering-11-00665-f014:**
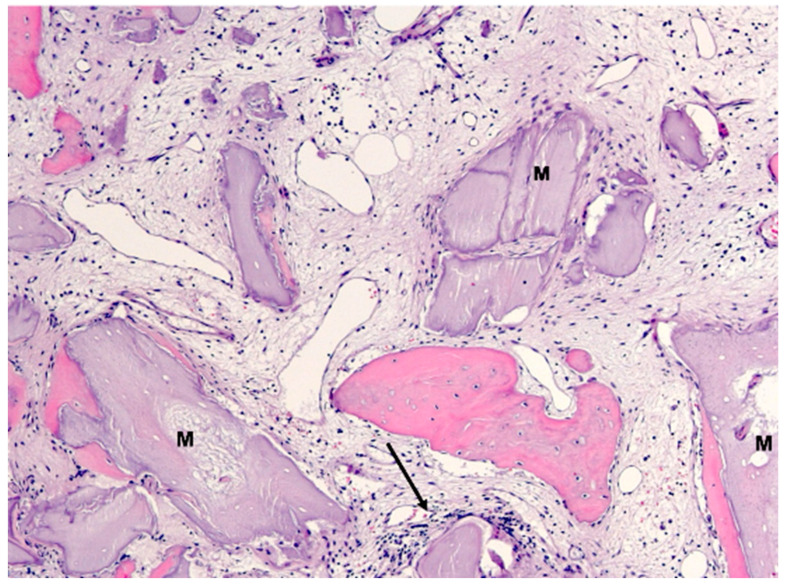
Biopsy case 3, bone graft granules (M) with no or moderate osteogenesis within loose connective tissue, arrow = infiltration, HE, original magnification, ×10.

**Figure 15 bioengineering-11-00665-f015:**
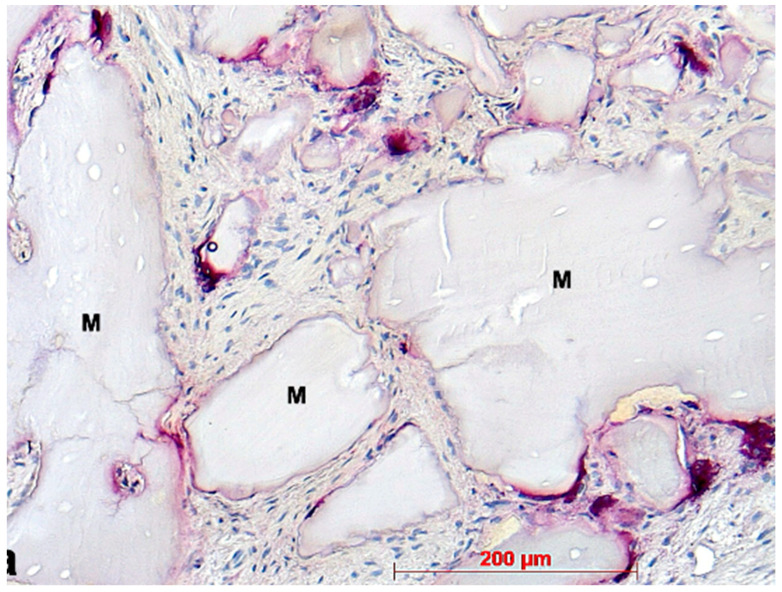
Identification of osteoclasts (purple stained cells): biopsy case 7, osteoclasts around bone graft granules (M), TRAP, original magnification ×20.

**Figure 16 bioengineering-11-00665-f016:**
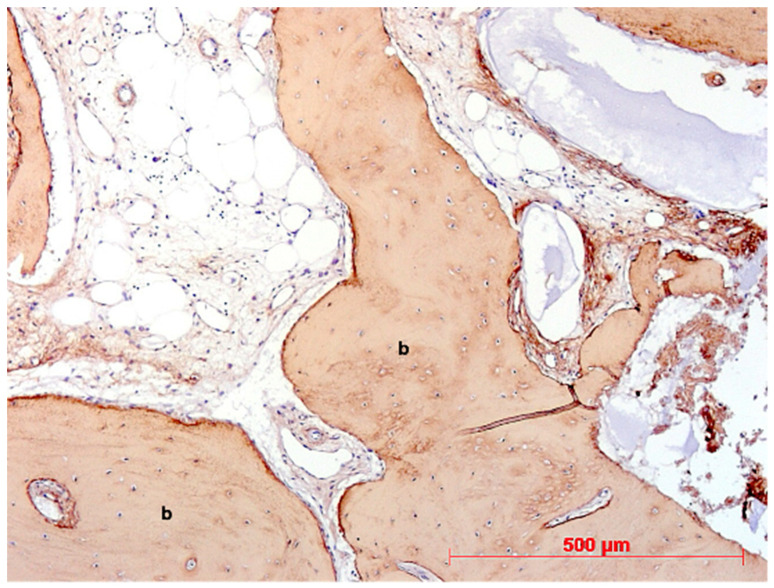
Biopsy case 2, collagen type I immunostaining (brown) in newly formed bone (b), DAB, original magnification ×10.

**Figure 17 bioengineering-11-00665-f017:**
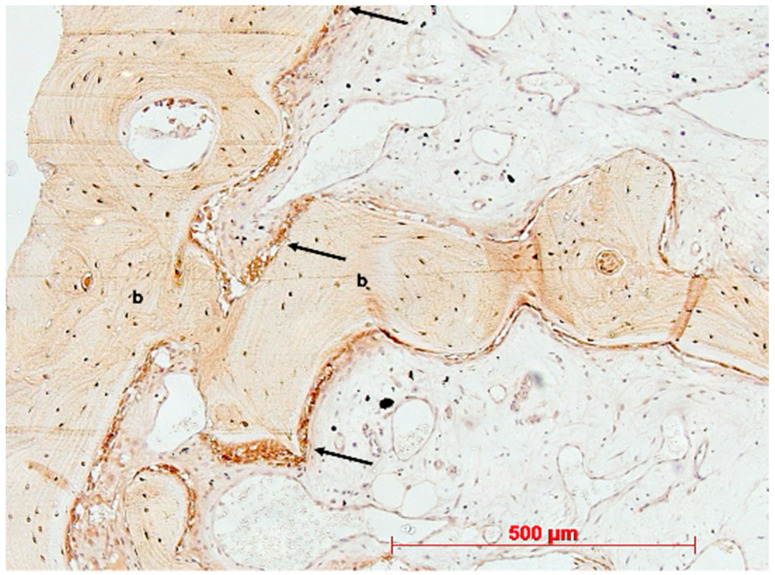
Biopsy case 3, osteocalcin immunostaining (brown) in newly formed bone (b) and osteoblasts (arrows), DAB, original magnification ×10.

**Figure 18 bioengineering-11-00665-f018:**
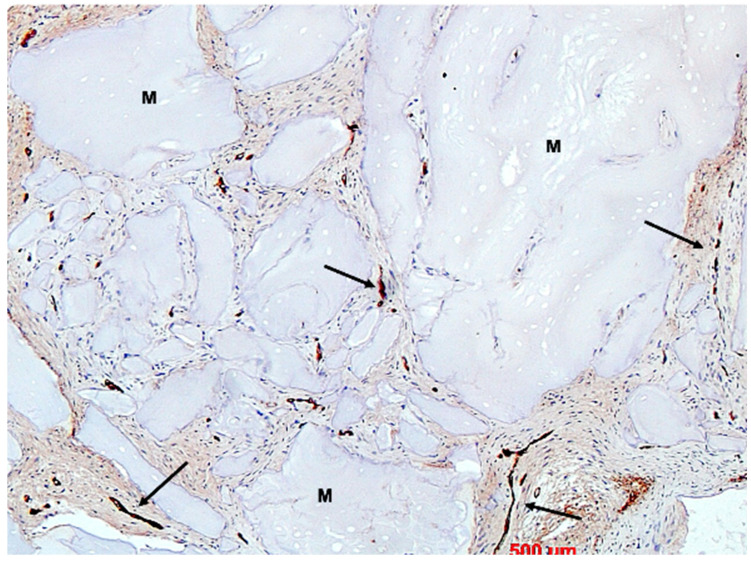
Biopsy case 3, vWF immunostaining for vessels (brown, arrows), M = bone graft granules, DAB, original magnification, ×10.

**Table 1 bioengineering-11-00665-t001:** Descriptive data related to the cases.

Case No.	Sex	Age	Regio (FDA):	Reason for Tooth Loss:	Integration Period of the Graft:	Smoking Behavior:	General Medical History:
1	m	42	25	fracture	130	never	no abnormal medical history
2	m	63	17	endodontic	319	10 cig./d	no abnormal medical history
3	m	45	25	unrestorable	133	never	no abnormal medical history
4	f	77	36	fracture	207	former smoker	type 2 diabetes, medicated
5	f	62	46	endodontic	190	never	no abnormal medical history
6	m	54	37	endodontic	253	former smoker	type 2 diabetes, medicated
7	m	55	37	tooth tilting	292	never	atopic individual
8	f	43	26	endodontic	187	former smoker	no abnormal medical history
9	m	45	25	unrestorable	146	former smoker	atopic individual
10	f	44	15	endodontic	184	never	no abnormal medical history

**Table 2 bioengineering-11-00665-t002:** Antibody details and incubation protocols; hp = heat pretreatment, on = overnight, rt = room temperature.

Antibody	Isotype	Manufacturer	Incubation Protocol
collagen type I	rabbit monoclonal	Abcam (Cambridge, UK)	1:400, 1 h, rt
osteocalcin	mouse monoclonal	Takara (Otsu, Shiga, Japan)	1:100, 1 h, rt
von Willebrand Factor (vWF)	rabbit polyclonal	Linaris (Wertheim, Germany)	1:200, 1 h, rt

## Data Availability

The data presented in this study are available on request from the corresponding author due to ethical reasons.

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
