# Peer review of "Alveolar Ridge Preservation Using a Novel Species-Specific Collagen-Enriched Deproteinized Bovine Bone Mineral: Histological Evaluation of a Prospective Case Series"

_bioengineering, 2024, doi:10.3390/bioengineering11070665_

Round 1

Reviewer 1 Report

Comments and Suggestions for Authors

Some improvements should be added to the paper, for example the histology section in materials and methods is completely missing. Figures do no report signs or letters as indicated in captions. Moreover a histomorphometric approach to calculate the amount of new bone would greatly improve the quality of the paper, on the other hand what is the purpose to study collagen I immunologically when  new bone is clearly visible in sections, of course there will be collagen I. The discussion could be wider, discussing also the histolological findings.

here are more specific comments:

line 139:  and eight patients were non-smokers.....there is a problem with numbers...these should be five non smokers.

line 199: Figure 3. Socket of tooth 46 after membrane placement.....is it the membrane or the C-DBBM?

The Materials and Methods section does not report anything on specimens processing (decalcification, embedding) and techniques used for histological evaluation, as staining, immunostainig etc..

Figure 8: scale bar is missing while is present in all other figures.

Figure 11: asterisks are missing in the figure, as well as the notation indicated in the caption (M = MOX granules, Md = degraded MOX granule)

Figure 12-15, 17, 18 same comment as above, no arrows, or asterisks or stars or letters on the figures while repoted in the caption

line 292:A clear correlation between the progression of osteogenesis and bone substitute laying time was not obvious. In the patients suffering from diabetes and atopism, osteogenesis was poorly developed.....This sentence would be better supported if a histomorphometrical analysis were done, i.e. calculate the amount of new bone in a central core section for each specimen

line 326: I would add the lack of quantitative data in terms of histomorphomtry as another limitation.

Author Response

Reviewer 1:

Some improvements should be added to the paper, for example the histology section in materials and methods is completely missing.

Agree. The requested section has been edited.

Figures do not report signs or letters as indicated in captions. Moreover a histomorphometric approach to calculate the amount of new bone would greatly improve the quality of the paper, on the other hand what is the purpose to study collagen I immunologically when new bone is clearly visible in sections, of course there will be collagen I. The discussion could be wider, discussing also the histolological findings.

Agree. We have significantly revised the Figures section. Immunohistochemistry for osteocalcin and collagen type I was used to evaluate the quality of newly formed bone and to demonstrate that this new material combination exhibits adequate clinical performance. We now plan to conduct a much larger randomized controlled trial (RCT) using the most well-studied bovine bone substitute material containing porcine collagen as a benchmark. This will allow us to compare it more definitively with the bovine bone substitute material investigated in this study. We aim to work with significantly greater standardization regarding the material's healing time, patient population, implant sites, histomorphometric approach etc. Due to the current lack of information on the bone substitute material investigated here, we have decided to present our current results at this stage. Expanding the discussion on the histological findings is somewhat problematic due to the small sample size and lack of standardization. Therefore, we prefer to take a cautious position regarding the generalization of the histological findings. We aim to conduct a larger, more standardized study within the framework of a randomized controlled trial (RCT). However, this work will likely require several more years. Hence, we believe that publishing this small case series is valuable to provide an initial insight.

here are more specific comments:

line 139:  and eight patients were non-smokers.....there is a problem with numbers...these should be five non smokers.

Agree. The requested section has been edited.

line 199: Figure 3. Socket of tooth 46 after membrane placement.....is it the membrane or the C-DBBM?

Agree. The requested section has been edited.

The Materials and Methods section does not report anything on specimens processing (decalcification, embedding) and techniques used for histological evaluation, as staining, immunostainig etc..

Agree. The requested section has been edited.

Figure 8: scale bar is missing while is present in all other figures.

Agree. The missing scale bar in Fig. 8 is due to the reconstruction of various individual images at 5x magnification, where we did not insert the individual scale bars to avoid them appearing as disruptive elements in the overall composition. We can either explain this, as the original magnification of 5x is indicated in the legend, or we can insert a scale bar. Both options are acceptable to us, but we are happy to comply with your preference. If you prefer the scale bar, we can provide it within a short time.

Figure 12-15, 17, 18 same comment as above, no arrows, or asterisks or stars or letters on the figures while repoted in the caption

I kindly apologize for the transmission error. A corrected version will be provided.

line 292: A clear correlation between the progression of osteogenesis and bone substitute laying time was not obvious. In the patients suffering from diabetes and atopism, osteogenesis was poorly developed.....This sentence would be better supported if a histomorphometrical analysis were done, i.e. calculate the amount of new bone in a central core section for each specimen

Agree. The requested section has been edited.

line 326: I would add the lack of quantitative data in terms of histomorphomtry as another limitation.

Agree. The requested section has been edited.

Reviewer 2 Report

Comments and Suggestions for Authors

The current version of the manuscript requires major corrections and additions. Suggestions for improving the manuscript are as follows:

1. In the abstract, it is not necessary to use headings. "We strongly encourage authors to use the following style of structured abstracts, but without headings" (https://www.mdpi.com/journal/bioengineering/instructions - Microsoft Word template)

2. Ten patients were included in this study. Is the sample size sufficient? Please elaborate in the manuscript. In my opinion, sample sizes must be many times larger. The question of the reliability of the results on such a small sample is raised.

3. The last paragraph of the Introduction section needs to be rewritten. It is not enough to write what the focus of the research is. First, the shortcomings of previous research should be highlighted, i.e. research gap. Then the scientific contribution and scientific hypotheses of this research should be written.

4. The scientific contribution of this research is currently hidden. The innovation of this research should be emphasized. Specifically, what is new in applied methods and/or experimental research?

5. The research methodology should be presented as universal, only after that comes the concretization that is currently shown in section 2.

6. Additionally, the authors in section 2 list the materials and methods but do not explain why something was selected. Every choice must be justified.

7. Descriptive data related to the cases are presented in Table 1. How does this data affect the results? Should be analysed and discussed in detail.

8. Detailed information about the bone graft should be displayed.

9. Potential errors were not analysed and discussed.

10. What is the uncertainty of the obtained results?

11. All figures must be analysed and discussed in more detail. Scientific discussion should be more intensive. The influence of input parameters on output parameters should also be discussed in detail.

12. The conclusions should state the benefits of this research, the possibilities of practical application and future research.

Author Response

The current version of the manuscript requires major corrections and additions. Suggestions for improving the manuscript are as follows:

  1. In the abstract, it is not necessary to use headings. "We strongly encourage authors to use the following style of structured abstracts, but without headings" (https://www.mdpi.com/journal/bioengineering/instructions - Microsoft Word template)

Agree. The requested section has been edited.

  1. Ten patients were included in this study. Is the sample size sufficient? Please elaborate in the manuscript. In my opinion, sample sizes must be many times larger. The question of the reliability of the results on such a small sample is raised.

Agree. The requested section has been edited.

  1. The last paragraph of the Introduction section needs to be rewritten. It is not enough to write what the focus of the research is. First, the shortcomings of previous research should be highlighted, i.e. research gap. Then the scientific contribution and scientific hypotheses of this research should be written.

Agree. The requested section has been edited.

  1. The scientific contribution of this research is currently hidden. The innovation of this research should be emphasized. Specifically, what is new in applied methods and/or experimental research?

Agree. The requested section has been edited.

  1. The research methodology should be presented as universal, only after that comes the concretization that is currently shown in section 2.

Agree. The requested section has been edited.

  1. Additionally, the authors in section 2 list the materials and methods but do not explain why something was selected. Every choice must be justified.

Agree. The requested section has been edited.

  1. Descriptive data related to the cases are presented in Table 1. How does this data affect the results? Should be analysed and discussed in detail.

Agree. The requested section has been edited.

  1. Detailed information about the bone graft should be displayed.

Agree. The requested section has been edited.

  1. Potential errors were not analysed and discussed.

Agree. The requested section has been edited.

  1. What is the uncertainty of the obtained results?

The requested section has been edited.

  1. All figures must be analysed and discussed in more detail. Scientific discussion should be more intensive. The influence of input parameters on output parameters should also be discussed in detail.

Agree. The requested section has been edited.

  1. The conclusions should state the benefits of this research, the possibilities of practical application and future research.

Agree. The requested section has been edited.

Round 2

Reviewer 1 Report

Comments and Suggestions for Authors

The paper has been greatly improved.

Please check line 341...Figure 13. Biopsy case 1, osteoblasts on the surface of newly formed bone (open arrows), stars = 341 fibrous bone, O = osteoclasts, PAS, original magnification, ×40.....

PAS is not mentioned in mat and met and is maybe a mistake

Author Response

Dear Reviewer,

Thank you very much. I greatly appreciate your thoroughness in identifying our error. We have corrected the mistake and updated the manuscript accordingly. Thank you for your meticulous review and valuable feedback.

Reviewer 2 Report

Comments and Suggestions for Authors

The manuscript has been corrected and improved.

Author Response

Thank you very much